# Research on the speed fluctuation control of diesel engine under load changes via an improved sparrow algorithm

Jun Fu [1,2,3]*, Luchen Lin[1], Shuo Gu[1], Han He[1], Zhenghong Chen[4]

**1** College of Mechanical and Energy Engineering, Shaoyang University, Shaoyang, China, **2** Key Laboratory of Hunan Province for Efficient Power System and Intelligent Manufacturing, Shaoyang University, Shaoyang, China, **3** Key Laboratory of Hunan Province for Advanced Agricultural Machinery Equipment and Key Manufacturing Technology, Shaoyang University, Shaoyang, China, **4** Hunan Binhu Diesel Engine Co., Ltd, Yiyang, Hunan, China

* 4160@hnsyu.edu.cn

## Abstract

Given the nonlinear and time-varying characteristics of diesel engine speed control, a conventional proportional integral derivative (PID) controller is inadequate for addressing the lag or overshoot in the system response, and it struggles to adapt to complex dynamic changes under load. This study proposes a fuzzy proportional integral derivative (FPID) control, which is based on an improved sparrow search algorithm(ISSA) with the aim of enhancing the system's adaptability. By refining the algorithm to augment its parameter control capabilities and employing test functions for experimental comparisons, the improved algorithm exhibited accelerated convergence and increased accuracy. The improved sparrow search algorithm is applied to two controllers for experimental comparison, and the results indicate that, in contrast to the traditional PID control algorithm, the FPID control algorithm reduces the adjustment time by 1.4 s and decreases the overshoot by 6.8% when the speed is adjusted to 2000 revolutions per minute (RPM). The duration for speed fluctuation stabilization under load changes of 8 and 10 is decreased by 18% and 30%, respectively, and the fluctuation deviation of the speed is reduced by 7% and 12%, respectively. Consequently, the implementation of FPID parameters tuned by the improved sparrow algorithm provides robust support for the stable operation of a diesel engine during speed fluctuations.

## 1 Introduction

The diesel engine is a complex machine because of its high thermal efficiency, ship, agricultural equipment, and generator main power. Speed is an extremely important parameter for diesel engines, and speed control is a particularly important part of diesel engine control. The speed control of diesel engines is affected by the high

**Data availability statement:** All relevant data are within the manuscript and its Supporting information files.

**Funding:** 1. Natural Science Foundation of Hunan Provincial, grant number 2022JJ50025. 2. Graduate Research Innovation Project of Hunan Provincial, grant number CX20240995.

**Competing interests:** The authors have declared that no competing interests exist.

nonlinearity and time-varying nature of diesel engines, and the change in state parameters under variable operating conditions has brought great challenges to the speed control of diesel engines [1–3].

A speed control system based on proportional integral derivative (PID) control is a common strategy for diesel engine speed control. PID controllers are widely used because of their simple structure, ease of implementation, and good control effects. However, the selection of parameters for PID controllers has a significant effect on the control performance [4,5]. Traditional PID controllers cannot perform online parameter tuning, and in complex systems such as diesel engines, which are nonlinear and time-varying, their control effects are often unsatisfactory [6]. In this context, artificial intelligence is currently used to optimize the parameters and structure of the PID [7–11] to enable the controller to meet complex control requirements, and popular types of intelligent PID controllers include fuzzy proportional integral derivative(FPID) [12,13]and neural network proportional integral derivative(NNPID) [14,15].

Boshun Zeng [16] proposed a speed control method that combines the improved salp algorithm with a PID controller to optimize the PID controller parameters of the diesel engine model through the improved Salp algorithm, aiming to improve the control accuracy and stability,during sudden changes in speed and load, the overshoot decreases by an average of more than 30.3% and more than 8.6%, respectively. Zhu [17] used the genetic Algorithm to regulate diesel engine fuel injection, and the results showed that the control performance improved after the algorithm was tuned. Hu [18] improved the particle swarm optimization(PSO) algorithm and applied it to dual-fuel engine control to enhance speed control stability. The response time was reduced by 0.47 s and the maximum overshoot was reduced by 98.43%. The above literature shows that the use of intelligent optimization algorithms for solving the controller parameters can provide good control results.

Because disturbances and uncertainties in real-world models are difficult to estimate, an adaptive fuzzy mechanism is used to estimate the unknown uncertainties and external disturbances. Ding et al. [19] introduced a parameter self-learning-based speed anti-disturbance control algorithm that quickly compensates for the load to enhance the speed anti-disturbance capability. Di et al. [20] combined PID control with fuzzy control and proposed a parameter self-tuning FPID controller for diesel engine speed control,which reduced the speed change rate from 26.61% to 7.51%, and the adjustment duration of fuel injection from 4.91s to 2.07s, which effectively improved the stability of the system. Jie et al. [21]. applied adaptive control to a V2 engine and achieved better performance than traditional PID control, and the results showed that the feedforward adaptive controller could significantly improve the tracking time by about 40% under ramp loads. Liu [22] proposed an optimized FPID algorithm applied to a fast temperature control system for automotive batteries, and the results showed that the comprehensive performance of the FPID composite controller was better than that of the ordinary PID controller and basic fuzzy controller,the temperature control time was reduced by approximately 76%. Fuzzy control can better adjust the PID parameters in real time in response to environmental changes; however, when the corresponding fuzzy control rules are designed, the

database parameters are often selected on the basis of personal experience, and the performance of fuzzy controllers is closely related to the human factor, which leads to difficulty in optimizing system performance [23]. Therefore, an optimization algorithm is used to optimally adjust the proportionality factor and quantization factor of the FPID to overcome the influence of human factors in the fuzzy control design as much as possible so that the system can be controlled more accurately and faster.

In recent years, the sparse search algorithm (SSA), an emerging intelligent optimization algorithm [24], has performed well in a variety of optimization problems owing to its simplicity, efficiency, and good adaptability [25–27]. Zhang Zhihuo [28] improved the SSA and applied it to the control of a lower-limb rehabilitation robot, and the results showed that the ISSA-PID control significantly reduced the error of the hip and knee joints by 63.3% and 72.5%, respectively, and achieved accurate gait tracking. Liang [29] addressed the problem of parameter optimization of the traditional PID controller and proposed a control method based on the SSA. Compared with the traditional PID control method, the response speed of the proposed method is increased by 64.1% and the interference adjustment time is shortened by 52.3%. Zhang [30] improved the sparrow algorithm for an extreme learning machine (ELM) neural network to propose a method for the inversion of soil parameters for underground space development and the prediction of deformation of underground structures.The relative error of the horizontal displacement monitoring value of the ground wall is reduced by 10%. Xue [31] optimized the variational modal decomposition process of the improved sparrow search algorithm and the total number of stress rainflow cycles was reduced by 17.1%, and the actual damage was reduced by 7.8% compared with the results obtained by the traditional method, which effectively reflected the fatigue effect caused by stress and improved the efficiency of stress spectrum compilation.The above scholars, addressed the inherent defects of the pigeon-inspired algorithm, such as population initialization imbalance, slow convergence speed, and tendency to trapped in local optima, by proposing different solutions. In the diesel engine speed control system, the optimization accuracy of control parameters directly affects the speed regulation performance, imposing stricter requirements on the global search capability and convergence accuracy of the optimization algorithm.

Therefore, this study proposes an improved sparse search algorithm(ISSA) to address the problem of the algorithm easily falling into local optima due to the high precision requirements for parameter adjustment in diesel engine speed control and achieves stable control of the diesel engine speed under load changes by optimizing the proportion factor and quantization factor of the FPID.

## 2 Diesel engine model

This study focuses on small diesel engines by establishing a diesel engine sub-model (cylinder, intake and exhaust, and crankshaft power model) to accurately reflect the operating characteristics of the diesel engine. Through the experimental setup shown in the Fig 1, accurate operating data of the diesel engine can be obtained and the simulation model can be verified to ensure the model error accuracy is less than 10%, providing a reliable simulation model for subsequent algorithm development. The basic parameters of the diesel engine used in this study are shown in Table 1.

To simplify the complex working process of internal combustion engines and to more easily describe the changes in the work mass within a diesel engine, some basic assumptions need to be made, and the energy conservation equation, mass conservation equation, and equation of state of an ideal gas are used to relate the working process of a diesel engine [32].

Mass conservation equations:

$$\frac{dm}{d\varphi} = \frac{dm_s}{d\varphi} + \frac{dm_e}{d\varphi} + \frac{dm_b}{d\varphi}$$

(1)

where $m_s$ denotes the mass of the cylinder intake, $kg$, $m_e$ is the mass of the in-cylinder mixture, $kg$ and $m_b$ denotes the amount of fuel injected into the cylinder, $kg$. $d\varphi$ is the differential of the crankshaft angle

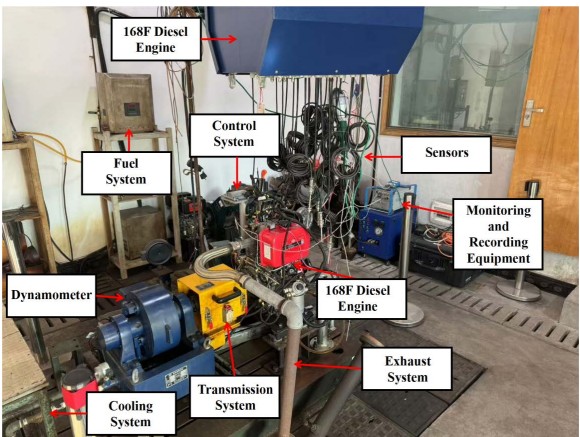

**Fig 1. Engine test bench.**

**Table 1. F168 technical parameters of a single-cylinder diesel engine.**

| Parameter | Unit | Value |
|---|---|---|
| Stroke | – | 4 |
| Number of cylinders | – | 1 |
| cylinder diameter | mm | 68 |
| Rated speed | r/min | 3300 RPM |
| Rated power | kW | 2.3 KW |
| Piston stroke | mm | 54 |
| Rod length | mm | 84 |
| Compression ratio | – | 22 |

Energy conservation equation:

$$\frac{dU}{d\varphi} = \frac{dQ_B}{d\varphi} + \frac{dm_s}{d\varphi}h_s - \frac{dm_e}{d\varphi}h_e - \frac{dQ_w}{d\varphi} - p\frac{dV}{d\varphi}$$

(2)

where $U$ is the internal energy of the system, $Q_B$ is the heat released by the combustion of fuel, $Q_W$ is the amount of heat exchanged with the outside world through the perimeter wall of the cylinder, $h_s$, $h_e$ is the specific enthalpy of the work mass at the inlet and exhaust valves, $p$ is the pressure of the work mass in the cylinder, $V$ is the working volume of the cylinder, and $m_s$, $m_e$ is the mass of inflow to and outflow from the cylinder.

Ideal gas state equation:

$$pV = mRT$$

(3)

where $V$ is the volume, $T$ is the temperature (K), $R$ is the gas constant, $kJ/(kg.K)$.

## 2.1 Cylinder model

The model of the cylinder module consists of three aspects: the cylinder working volume, combustion, and cylinder perimeter wall heat transfer. The module plays a vital role in capturing and modeling the complex dynamics occurring within the engine cylinder. The cylinder working volume is expressed as follows:

$$V_s = \frac{\Pi D^2}{4}\{\frac{S}{\varepsilon-1} + \frac{S}{2}[(1 + \frac{1}{\lambda}\sqrt{1 - \lambda^2\sin^2(\frac{\Pi}{180}\varphi)}]\}$$

(4)

where $V_S$ is the working volume of the cylinder, $L$, D is the cylinder diameter, mm; $\lambda$ is the connecting rod-to-crank ratio; $s$ is the cylinder stroke; $\varepsilon$ is the compression ratio; $\varphi$ is the cylinder diameter.

In the in-cylinder combustion exothermic law, the double Weber exothermic rate model is used to divide the entire combustion process into two parts: premixed combustion and diffusion combustion, with two Weber function curves used to indicate premixed combustion and diffusion combustion; the two in the calculation are forked off from a certain leading angle and then the two are superposed, integrating the expression of the in-cylinder combustion exothermic rate:

$$X = X_1 + X_2$$

(5)

$$\frac{dx}{d\varphi} = \frac{dx_1}{d\varphi} + \frac{dx_2}{d\varphi}$$

(6)

where $X_1, X_2$ are the fuel fractions of premixed and diffusion combustion, respectively.

According to the basic formula of heat transfer, the amount of heat transferred from the working fluid in the cylinder to the cylinder wall is as follows:

$$\frac{dQ_w}{d\varphi} = \sum_{i=1}^{3}\frac{dQ_{wi}}{d\varphi} = \frac{1}{w}\sum_{i=1}^{3}\alpha_g \cdot A_i (T - T_{wi})$$

(7)

where $a_g$ is the instantaneous average heat transfer coefficient; $A_i$ is the surface area of the heat transfer surface, $m^2$, T is the temperature of the work material in the cylinder, K; and $T_{wi}$ is the average temperature of the wall, $K$, where $i = 1, 2, 3$, are the three parts of the circumference wall of the cylinder, cylinder head, piston, and cylinder liner, respectively.

The instantaneous average heat transfer coefficient $\alpha_g$ is expressed via the Woschni formula:

$$\alpha_g = 130 \cdot D^{0.2} \cdot p^{0.8} \cdot T^{-0.53}\left[C_1 \cdot C_m + C_2 \cdot \frac{V_s \cdot T_{cj}}{P_{cj} \cdot V_{cj}}(p - p_{c0})\right]^{0.8}$$

(8)

where $p$ is the cylinder mass pressure (bar), $D$ is the cylinder diameter, $m$, $T_{cj}$ is the cylinder mass temperature at the beginning of compression (K); $V_S$ is the average piston speed, $m/s$, and $p_{c0}$ is the cylinder pressure at the start time (bar), where $C_u$ is the inlet vortex speed, $m/s$.

## 2.2 Intake and exhaust model

The flow calculations for the intake and exhaust valves included the rate of change of flow through the valve into and out of the cylinder, the geometric cross-sectional area of the flow, and the flow coefficient. In a unit crankshaft angle, the flow rate dm through the valve is expressed as follows:

$$dm = \frac{1}{w} \cdot \mu F\sqrt{\frac{p_1}{v_1}} \cdot \psi d\varphi$$

(9)

where $\omega$ is the angular velocity of the engine rotor shaft rotation, $rad/s$, µ is the flow coefficient; $F$ is the effective flow cross-sectional area, $m^2$, $\psi$ is the flow function; $p_1$ is the prevalence mass pressure (bar); and $v_1$ is the prevalent mass-specific volume.

## 2.3 Crankshaft dynamics model

In accordance with Newton's second law, all the torques acting on the diesel engine, including the indicated torque generated by fuel combustion, the friction torque, and the rate of change in the rotational speed of the engine load, are combined.

$$\frac{dn}{dt} = \frac{60}{2\Pi} \frac{M_i - M_f - M_L}{J_e}$$

(10)

where $J_e$ is the moment of inertia of the diesel engine, $M_i$ is the indicated torque in $N \cdot M$, $M_f$ is the friction torque in $N \cdot M$, and $M_L$ is the external load, $N \cdot M$.

## 2.4 Model validation

By comparing the predicted power value of the model with the actual diesel engine test power data and verifying the simulation accuracy of the model in terms of fuel injection and combustion efficiency, the accuracy of the diesel engine model can be effectively evaluated in terms of power output and fuel consumption rate simulation, and the response to the operating characteristics of the diesel engine can provide a basis for the design of subsequent algorithms. At the same speed, the power, fuel consumption rates and cylinder pressure of the simulation model and test bench data under loads of 25%, 50%, 75%, and 100% for the diesel engine were compared, as shown in Figs 2–4, where the gray line represents the test bench data and the red line represents the simulation results.

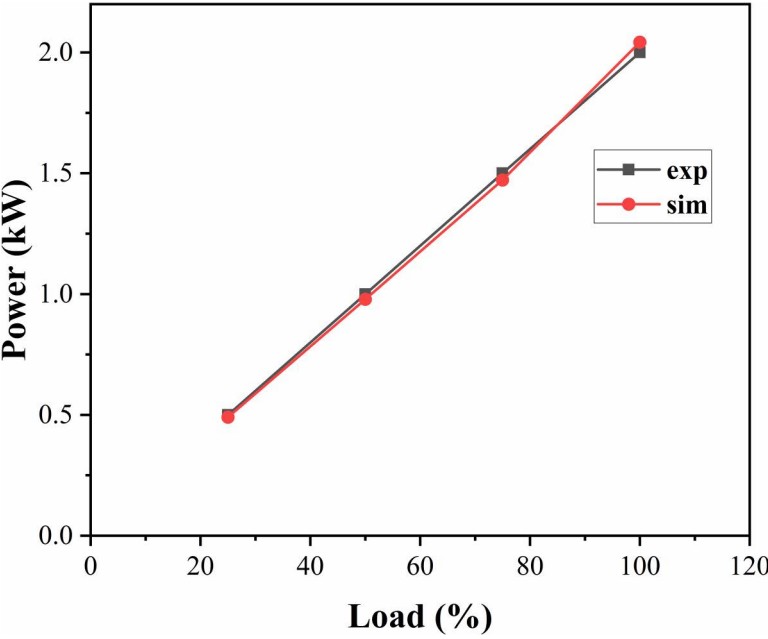

**Fig 2. Power curves under different loads.**

**Fig 3. Fuel consumption rate curve under different loads.**

According to the results of the comparative analysis, this diesel engine model is consistent with the actual data in terms of power output and can reflect the relationship between diesel engine speed and torque more accurately. The error in the fuel consumption rate is less than 10%, which indicates that the model has high accuracy in predicting fuel consumption. Under different loads, the actual cylinder pressure is basically consistent with the simulation model, and the peak pressure error is less than 5%. Therefore, the model effectively reproduces the combustion characteristics in the cylinder of a diesel engine and has a reasonable basis as a control model.

## 3 Fuzzy PID control with the improved sparrow algorithm

Through model verification, it is proven that the dynamic characteristics of the diesel engine are highly consistent with the simulation results. On this basis, this section proposes a fuzzy PID control method based on an improved sparrow search algorithm to address the limitations of traditional PID control in nonlinear systems [33].

### 3.1 Fuzzy PID

As shown in Fig 5, fuzzy logic is used to adjust the PID controller to control the speed of the diesel engine.

In a diesel engine, a Hall speed sensor is used to measure the flywheel speed and compare it with the set speed to obtain the rate of change in error E and error EC. These values were used as inputs for the FPID controller. Fuzzy rules are formed in fuzzy systems on the basis of fuzzy reasoning [34]. The three output parameters are defuzzified to obtain the final values, which are fed into the PID controller of the diesel engine control actuator. In this study, a simplified two-dimensional fuzzy controller was used to control the diesel engine speed system. The fuzzy controller considers the error $E$ and the rate of change of the error $EC$ as the input quantities and the control quantities as the output quantities. The inputs and outputs of the fuzzy controller are expressed in five levels {*negative large*, *negative small*, *zero*, *positive small*, *positive large*}, which are denoted by the letters {*NB*, *NS*, *Z*, *PS*, *PB*}, and the domain range of the inputs $E$ and $EC$ is $[-1, 1]$. The prepared fuzzy control rules based on control experience are shown in Fig 6 and Table 2.

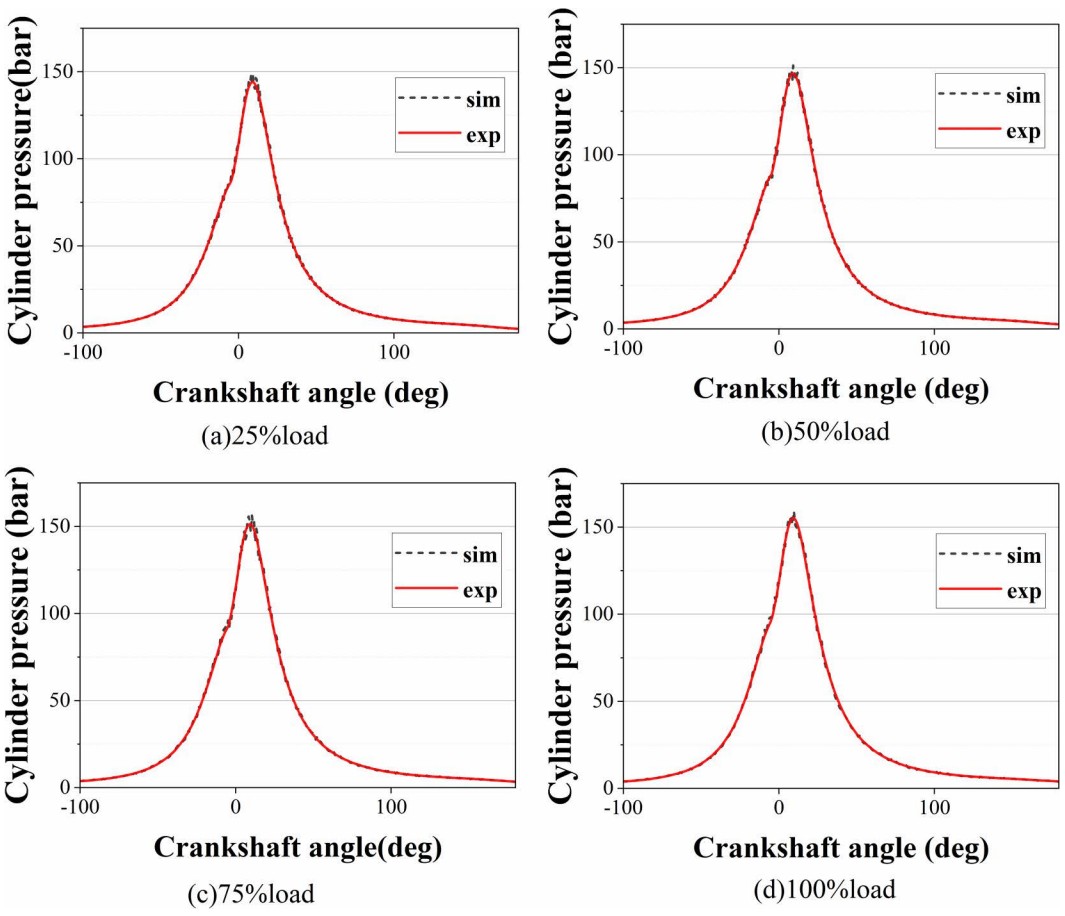

**Fig 4. Cylinder pressure under different loads.**

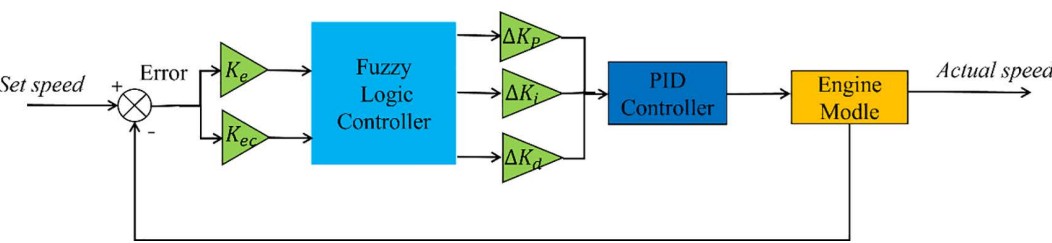

**Fig 5. Diesel engine control block diagram.**

**3.1.1 Objective function.** The focus of this study is on reducing the speed and speed deviation of a diesel engine to zero. The adaptation function $J$ time integral multiplied by the absolute error performance index used in this study is used to evaluate the control model via the RPM deviation value $E$ as the main base.

$$J = \int_0^\infty t|\Delta E|dt$$

(11)

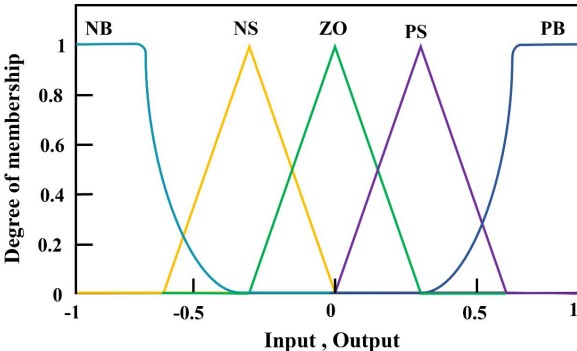

**Fig 6. Input, output membership function.**

**Table 2. Table of fuzzy rules.**

| E/EC | NB | NS | ZO | PS | PB |
|------|-----|-----|-----|-----|-----|
| NB | NB | NB | NS | NS | ZO |
| NS | NB | NS | NS | ZO | PS |
| ZO | NS | NS | ZO | PM | PS |
| PM | NS | ZO | PM | PM | PB |
| PB | ZO | PM | PM | PB | PB |

## 4 Optimization algorithms for fuzzy controllers

In the design and application of fuzzy controllers, their performance is significantly limited by human factors. From the formulation of fuzzy rules to the selection of proportional and quantization factors, all of them rely on human experience judgment, which makes it difficult to reach optimal system performance. To effectively overcome the inherent limitations of human factors in the fuzzy control design process, an optimization algorithm is introduced to optimize the proportionality and quantization factors of the fuzzy PID to determine the optimal parameter combinations and improve the performance of the fuzzy PID controller.

### 4.1 Sparrow search algorithm

The sparrow population can be abstracted into an explorer-follower-early warning model on the basis of the foraging and antipredation behaviors of the sparrow population. The sparrow algorithm was used to optimize the problem. If n sparrows exist in the d-dimensional search space, the position of the sparrow is expressed in a matrix as

$$X = \begin{bmatrix} x_{1,1} & \dots & \dots & x_{1,d} \\ x_{2,1} & \dots & \dots & x_{2,d} \\ \dots & \dots & \dots & \dots \\ x_{n,1} & \dots & \dots & x_{n,d} \end{bmatrix} \tag{12}$$

The corresponding fitness value is as follows:

$$F_X = \begin{bmatrix} f(x_{1,1} & \dots & x_{1,d}) \\ f(x_{2,1} & \dots & x_{2,d}) \\ f(\dots & \dots & \dots) \\ f(x_{n,1} & \dots & x_{n,d}) \end{bmatrix} \tag{13}$$

where d represents the dimension of the design variable of the problem to be optimized and n represents the number of sparrows.

In the search process, the explorer with a good fitness value is the first to obtain the food, and the finder has the largest search range.

$$X_{i,j}^{t+1} = \begin{cases} X_{i,j}^t \cdot exp(-\frac{1}{\alpha \cdot iter_{max}} \{ & R_2 < ST \\ X_{i,j}^t + Q \cdot L & R_2 \geq ST \end{cases}$$

(14)

where $t$ represents the number of iterations; $X_{i,j}^{t+1}$ indicates the value of the i row and j dimension at iteration t; $iter_{max}$ indicates the maximum number of iterations; $Q$ represents a random number that obeys a normal distribution; $L$ represents as a matrix of $1 \times d$, where each element is 1. $R_2$ and $ST$ are safety values and warning values, respectively. When $R_2 < ST$, sparrows perform global search foraging; when $R_2 \geq ST$, sparrows make random walks in a normal distribution.

The subscriber's location update formula is as follows:

$$X_{i,j}^{t+1} = \begin{cases} Q \cdot exp(\frac{X_{worst}^t - X_{i,j}^t}{t^2}) & i > n/2 \\ X_P^{t+1} + |X_{i,j}^t - X_P^{t+1}| \cdot A^+ \cdot L & otherwise \end{cases}$$

(15)

where $n$ represents the number of participants. At that time, n/2, sparrows with low fitness did not obtain food, so they needed to obtain more food. In other cases, sparrows randomly look for a location near the current optimal location.

The location of the early warning location update formula is as follows:

$$X_{i,j}^{t+1} = \begin{cases} X_{worst}^t + \beta \cdot |X_{i,j}^t - X_{best}^t| & if\ f_i \neq f_g \\ X_{i,j}^t + K(\frac{X_{i,j}^t - X_{worst}^t}{(f_i - f_w + \varepsilon)}) & if\ f_i = f_g \end{cases}$$

(16)

where $X_{worst}^t(X_{best}^t)$ is the worst (excellent) individual in the iteration, $f_i$ is the fitness value of the current individual, and $f_g$ is the current maximum fitness value.

When $f_i \neq f$ sparrows, it is on the edge of the population and is prone to being attacked by foragers. At this time, it flies toward the optimal sparrow. When $f_i = f$, the sparrow is at a dangerous position that is far from the worst sparrow position.

## 4.2 Improving the sparrow algorithm

Although the SSA possesses strong global search capabilities, it has an uneven distribution in high-dimensional complex parameter spaces, a slow convergence speed, and risks of falling into local optima, which may lead to non-globally optimal control parameter optimization results, thereby affecting the accuracy and stability of diesel engine speed control. To address these issues, this paper proposes three combined optimization strategies to improve the aforementioned shortcomings: introducing an improved Circle chaotic map to enhance the random distribution of the initial population and avoid population unevenness; adopting a dynamic factor during the exploration phase to initially expand and then progressively narrow the search range; utilizing a sine-cosine strategy that leverages the periodicity, fluctuation, and symmetry of sine and cosine functions to balance the algorithm's global search and local search capabilities; and employing a Levy flight strategy that combines long-step jumps with frequent short-step movements, enabling the search process to achieve both local refined exploration and globally efficient search.

### 4.2.1 Circle chaotic map.
To address the issue of uneven spatial distribution at the beginning of algorithm initialization, a circle chaotic mapping initialization method was proposed. The initial sparrow population is generated via circular chaotic mapping, which increases the diversity of the population and makes its distribution more uniform.

However, the original chaotic mapping is relatively dense in the range of [0.2, 0.6]; therefore, the formula is improved to make its chaotic values more uniform. The expression for the circle chaotic mapping is as follows [35]:

$$x_{t+1} = mod\left[x_t + 0.2 - (\frac{0.5}{2\pi})sin(2\pi x), 1\right]$$

(17)

The improved circle chaotic mapping expression is as follows:

$$x_{t+1} = mod\left[3.85x_t + 0.4 - (\frac{0.5}{3.85\pi})sin(3.85\pi x_t), 1\right]$$

(18)

When the dimension is 1000, a comparison of the histograms of the initial solutions of the circle chaotic mapping in Fig 7 reveals that after improvement, the distribution of the circle chaotic mapping is more uniform.

**4.2.2 Dynamic search factors.** The fixed search step size of the sparrow search algorithm is suitable for broad searches in the early stages, but it is not conducive to improving the convergence speed and accuracy of the algorithm in the later stages. Therefore, a dynamic search factor was added at the position of the producer to expand the search range in the early stages of the algorithm and gradually narrow it after each iteration to improve the accuracy of the algorithm. The formula for the dynamic search factor is

$$\alpha = 2\left(1 - \frac{T}{M}\right) + 0.1$$

(19)

where $T$ is the current iteration and where $M$ is the total number of iterations.

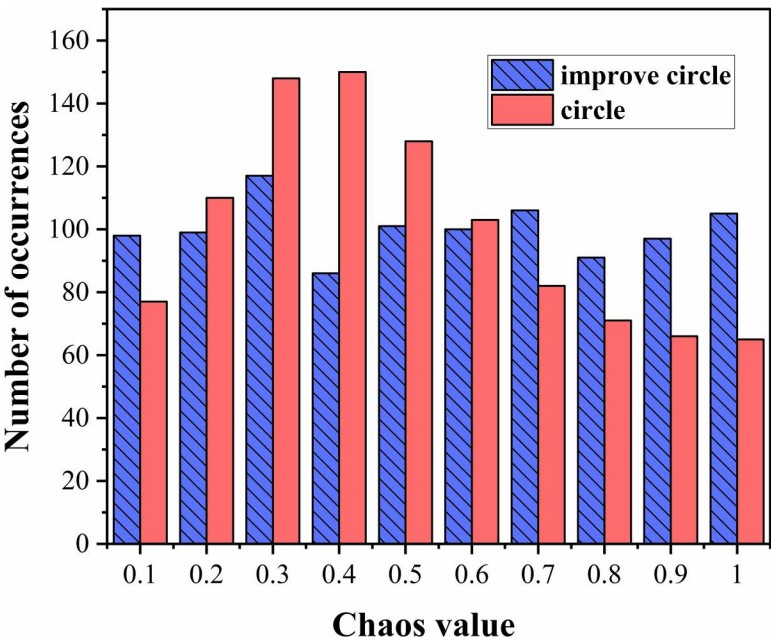

**Fig 7. Comparison of the distribution histograms of circular chaotic mapping.**

The new producer update formula is as follows:

$$X_{i,j}^{t+1} = \begin{cases} \alpha \cdot Q \cdot \exp(\frac{X_{worst}^t - X_{i,j}^t}{t^2}) & i > n/2 \\ X_P^{t+1} + \alpha \cdot |X_{i,j}^t - X_P^{t+1}| \cdot A^+ \cdot L & \text{otherwise} \end{cases} \quad (20)$$

**4.2.3 Sine and cosine strategies.** When the food searched by a finder is located in a local optimum, many followers flock to the location, causing a loss of population location diversity, which in turn increases the likelihood of falling into a local extreme. By using the oscillatory variation property of the positive cosine model to act on the position of the discoverer, the diversity of individual discoverers is maintained, which in turn improves the global search ability. The new subscriber's location update formula is

$$X_{i,j}^{t+1} = \begin{cases} X_{i,j}^t + r_2 \mathrm{Sin}(r_3)\left|r_4 X_{best}^t - X_{i,j}^t\right| & r_1 < 0.5 \\ X_{i,j}^t + r_2 \mathrm{Cos}(r_3)\left|r_4 X_{best}^t - X_{i,j}^t\right| & \text{otherwise} \end{cases} \quad (21)$$

where $r_1, r_2, r_3, r_4$ are random numbers; $r_2$ determines the movement distance of the sparrow; and $r_3$ controls the effect of the optimal individual on the sparrow's latter position.

**4.2.4 Levy flight strategy.** The addition of the levy flight strategy at the position of the alarmist ensures that the algorithm has good global and local search abilities, reduces the probability of falling into a local optimum, and improves the accuracy of the optimization search. The global search formula after adding the levy flight strategy is as follows:

$$X_{i,j}^{t+1} = \begin{cases} X_{worst}^t + \beta \cdot |X_{i,j}^t - X_{best}^t| \cdot levy(d) & \text{if } f_i > f_g \\ X_{i,j}^t + K(\frac{X_{i,j}^t - X_{worst}^t}{(f_i - f_w + \varepsilon)}) & \text{if } f_i = f_g \end{cases} \quad (22)$$

The pseudo-code implementation process of improving the sparrow algorithm is as follows:

## Algorithm 1 The Improving the sparrow algorithm(ISSA)

```
Input:
n:the number of sparrows
PD:the number of the producer sparrows
SD:the number of the sparrows which perceive danger
Iter_max:the total number of iterations
R_2:the alarm value
Output:
X_best:the global optimal position
f_best:the optimal solution
1:Initialize a population of n sparrows by(18)
2:whiel(t<Iter_max)
3:  Calculating the fitness values of the individuals;
4:  Ranking the fitness values and finding the current best individual and the current worst
individual;
5:    For i=1: PD do
6:Using (20) to update the producer position;
7:   end for
8:   For i = (PD+1):n do
9:    Using (21) to update the scrounger position;
10:   end for
11:   For i=1: SD do
```

```
12:    Using (22)to update the Scout Position;
13:    end for
14:    Obtaining the current new position;
15:    if (the new position is better than the best position)
16:    Replacing the best position with the new position
17:    end if
18:    t=t+1;
19:    end while
20:    return X_best, f_best;
```

## 4.3 Performance testing

To verify the performance and optimization results of the improved SSA, four benchmark functions with significant optimization characteristics were selected. These functions were independently run 30 times to avoid the influence of random numbers, with 1000 iterations set and a population size of 50. The specific dimensions, ranges, and extreme values of the test functions are detailed in Table 2.

In this study, four optimization algorithms were compared, namely, particle swarm optimization (PSO), gray wolf optimization (GWO), the Sparrow algorithm (SSA), and the improved Sparrow algorithm (ISSA). Fig 8 and Table 3 show the convergence curves of these algorithms on the four test functions, from which it can be seen that the ISSA algorithm exhibits the best performance among all test functions. Table 4 shows the average values after 30 tests of the algorithm, demonstrating the high stability and accuracy of the ISSA algorithm compared with the four other algorithms. The results

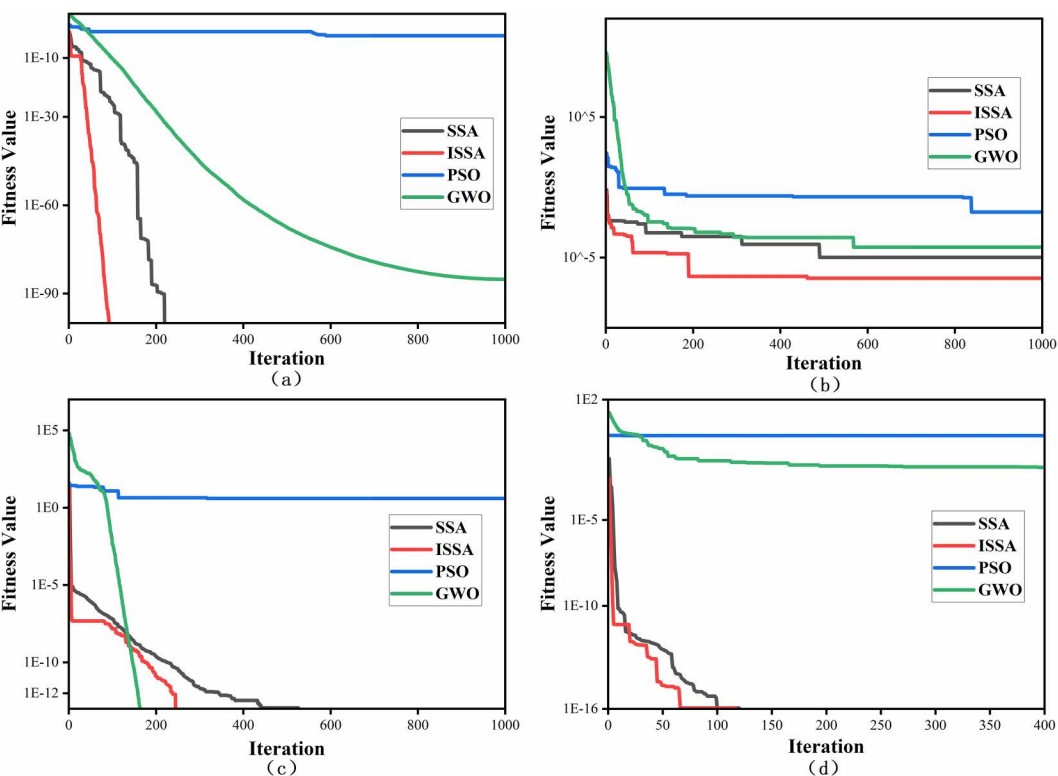

**Fig 8. Convergence plots of the benchmark experiments.** (a) f1 (b) f2 (c) f3 (d) f4.

**Table 3. Test functions.**

| Function | Dimension | Calculation Formula | Interval | Extremum |
|---|---|---|---|---|
| $f_1$ | 30 | $$f_1 = \sum_{i=1}^{n} x_i^2$$ | $[-100, 100]$ | 0 |
| $f_2$ | 30 | $$f_2 = \sum_{i=1}^{n-1} i x_i^4 + random[0, 1)$$ | $[-100, 100]$ | 0 |
| $f_3$ | 30 | $$f_3 = \sum_{i=1}^{n} \left[ (x_i^2 - 10cos(2\pi x_i))^2 + 10 \right]$$ | $[-512, 512]$ | 0 |
| $f_4$ | 30 | $$f_4 = \frac{1}{4000} \sum_{i=1}^{n} x_i^2 - \prod_{i=1}^{n} cos(\frac{x_i}{\sqrt{i}}) + 1$$ | $[-100, 100]$ | 0 |

**Table 4. Comparison of test results of different algorithms.**

| Indicator | Function | ISSA | SSA | GWO | PSO |
|---|---|---|---|---|---|
| Beast value | F1 | 0 | 4.82E−98 | 5.86E-86 | 3.91E-3 |
| | F2 | 3.31E-07 | 1.03E-05 | 5.42E-05 | 0.017 |
| | F3 | 0 | 0 | 0 | 3.97 |
| | F4 | 0 | 0 | 0.01 | 0.17 |
| Average values | F1 | 0 | 2.5E-52 | 8.07E-85 | 0.047 |
| | F2 | 6.85E-06 | 4.16E-04 | 2.84E-04 | 0.104 |
| | F3 | 0 | 1.51E-14 | 0 | 7.721 |
| | F4 | 0 | 1.87E-16 | 0.023 | 0.154 |
| Strand deviation | F1 | 0 | 1.38E-51 | 1.6E-84 | 0.057 |
| | F2 | 6.14E-06 | 4.83E-04 | 1.57E-04 | 0.067 |
| | F3 | 0 | 4.70E-14 | 0 | 1.919 |
| | F4 | 0 | 1.87E-16 | 0.023 | 0.154 |

indicate that the ISSA algorithm exhibits stronger stability among the four test functions, suggesting that the algorithm can overcome local optimum issues and obtain the best test results.

In summary, the improved sparrow algorithm (ISSA) outperforms the other three algorithms on various test functions, showing significant advantages in terms of convergence speed and accuracy. Therefore, using the ISSA algorithm to solve the fuzzy PID control of diesel engines can yield the optimal control parameters. The solution process is illustrated in Fig 9.

## 5 Results and analysis

The ISSA algorithm is applied to the diesel engine control model, and the same control strategy is used to optimize the control parameters through the ISSA algorithm. The experiments that followed were all simulations. As shown in Fig 10, the comparison between the SSA and ISSA algorithms reveals that the ISSA achieves a smaller iteration error after only 17 iterations and obtains a global optimal solution, demonstrating a strong rapid convergence capability. This indicates that the additional computation time required by the ISSA algorithm is minimal, making it suitable for online optimization.

The control curve of the diesel engine is shown in Fig 9 when it is boosted from idle to 2000 RPM under the no-load condition. Fig 11 and Table 4 show that after the parameter optimization of the two controllers (FPID and PID controller), the FPID controller yields superior control results compared with those of the FPID controller. The FPID controller exhibited a faster response and could quickly reach the target speed within 1.66 seconds Table 5.

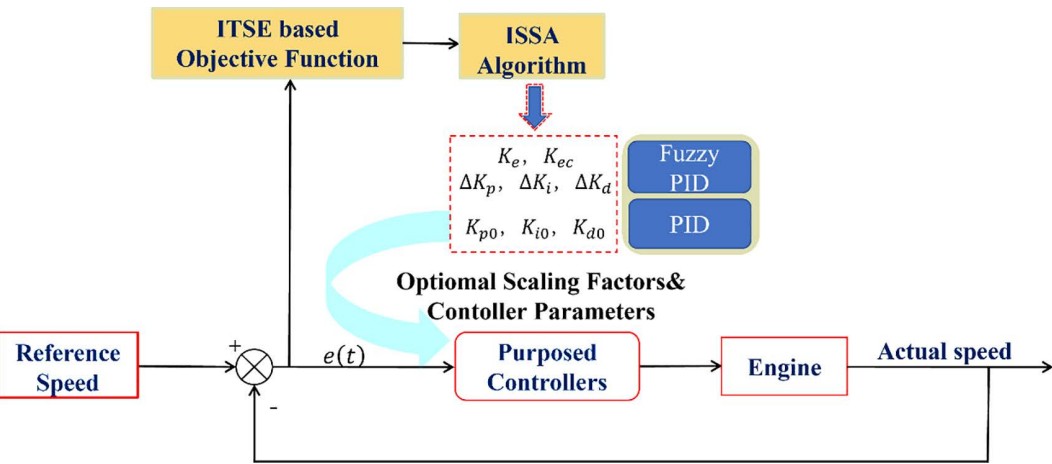

**Fig 9. Comprehensive control block diagram.**

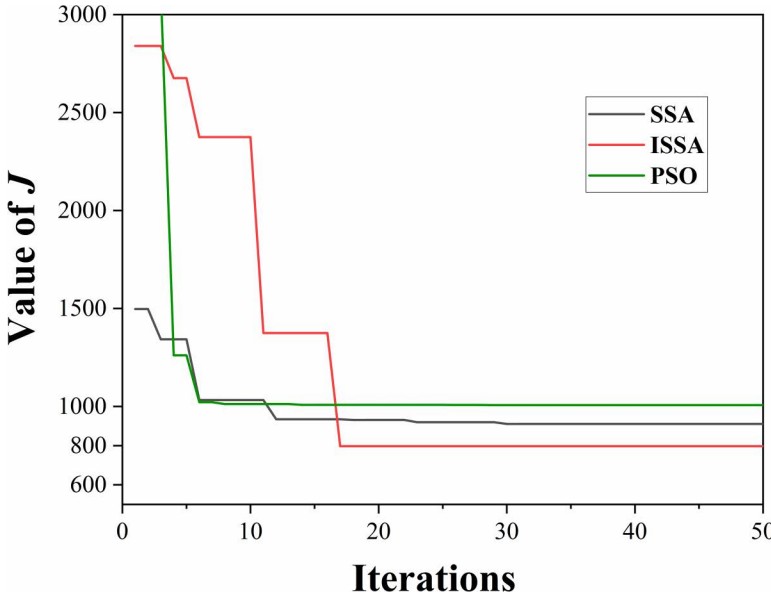

**Fig 10. Variation of the fitness function.**

The effect of the load variation on the system performance was further analyzed when the diesel engine was running steadily at 1800 RPM. In the experiments, different loads were applied to the diesel engine under PID and FPID control, and the stability of the two control algorithms under load variation was observed. As shown in Fig 12, different loads are applied to the diesel engine. Fig 13 shows the curve of the diesel engine speed changing with the load. In the case of small load changes (load change at 5 s), the diesel engine speed fluctuates slightly, and both the PID and FPID control algorithms demonstrate good control performance. However, when the load increased to $8\ N \cdot m$ and $10\ N \cdot m$, the FPID control showed better stability and faster adjustment response time. When the load was increased to $8\ N \cdot m$, the maximum deviation in speed was $23\ RPM$ with FPID control compared with $33\ RPM$ with PID control; when the load was restored to its original value (11th second), the adjustment time was 0.5 seconds faster with FPID control than with PID

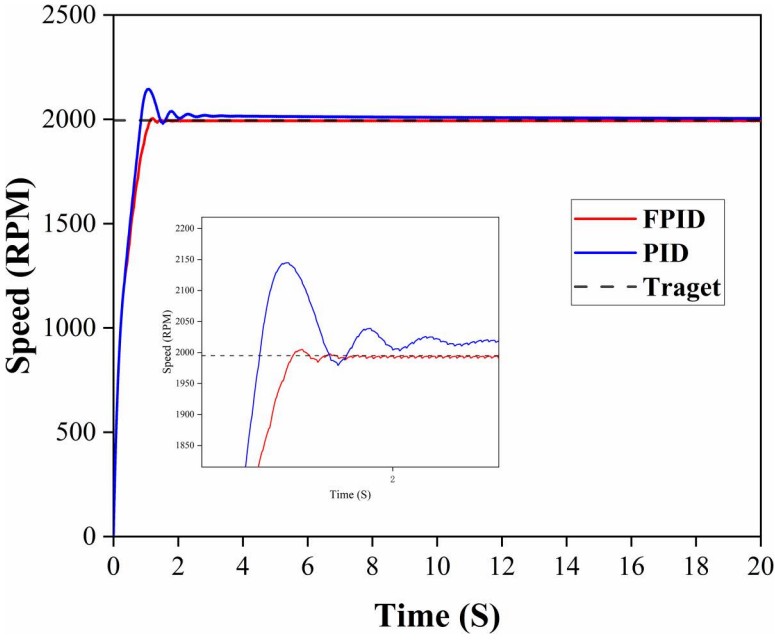

**Fig 11. The response curve at the idle state is raised to 2000 RPM.**

**Table 5. Comparison of Controller Effects at 2000 RPM.**

| Control Algorithm | Overshoot/% | settling time/s |
| --- | --- | --- |
| ISSA_PID | 7.2 | 3.06 |
| ISSA_FUZZY_PID | 0.4 | 1.66 |

control. When the load was increased to $10\ N \cdot m$, the difference between the FPID control and PID control became more obvious. At this point, the maximum deviation of the speed is $40\ RPM$ under FPID control compared with $60\ RPM$ under PID control; after the load is restored at the 17th second, the speed adjustment time for FPID control is reduced by $0.7\,s$ compared with that of PID control.

Through the above experiments, it is proven that FPID outperforms PID in speed regulation of diesel engines without load: FPID shows better stability and faster regulation speed in the case of large load fluctuations when the diesel engine is working stably.

## 6 Conclusion

This study addresses the issue of speed fluctuations in small-power diesel engines under varying loads and proposes a method for tuning the FPID controller in a diesel engine control system on the basis of an improved sparrow algorithm (ISSA). (1) The original sparrow algorithm is improved by homogenizing the initial parameters of the sparrow algorithm and introducing random factors, a positive cosine strategy, and levy flight strategy operations; thus, a more efficient ISSA algorithm is proposed. The simulation results show that, compared with the traditional SSA, the ISSA has significant advantages in terms of both convergence speed and convergence accuracy. (2) In this study, an FPID controller was designed, in which fuzzy logic was used to adjust the parameters of the PID controller adaptively. A robustness test was conducted to verify the performance of the controller. The test results show that the FPID controller optimized by the ISSA

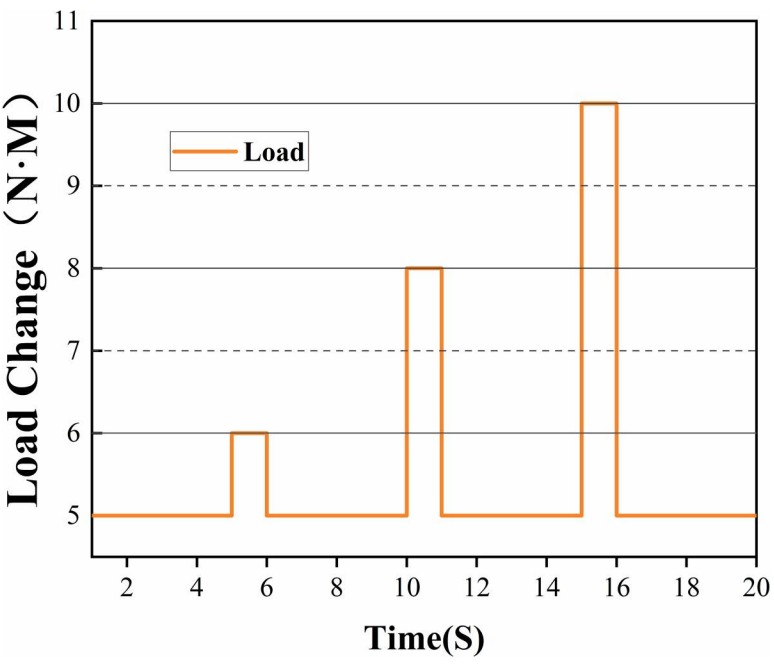

**Fig 12. Different load 6-10N·M.**

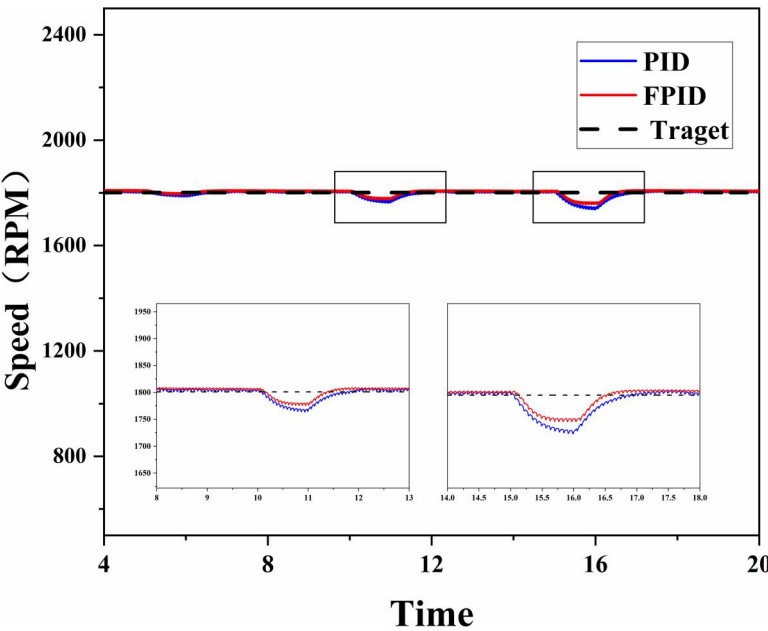

**Fig 13. Response curve under load variation.**

algorithm is more robust and has a better control effect under load variation than the traditional PID controller regulated by the ISSA algorithm. In summary, the control strategy proposed in this paper effectively improves the dynamic performance and robustness of the control system and has better control accuracy and response speed in the rotational speed fluctuation under the diesel engine load change compared with PID. However, while the improved algorithm enhance the control effect, it also increase the compute, especially in the simulation and experimental processes, which reflects a higher computational demand and fuzzy control in the sudden change in load. Although it can adapt well to sudden changes in the diesel engine load, it still produces a certain amount of error. In the future, further research can be carried out on three aspects: algorithm efficiency improvement, hybrid control strategy design, and practical application verification.

## Supporting information

**S1 File. Engine.**
(XLSX)

**S2 File. Fitness.**
(XLSX)

**S3 File. Fitness F1-F4.**
(XLSX)

## Acknowledgments

The authors declare that have no known competing financial interests or personal relationships that could have appeared to influence the work reported in this paper.

## Author contributions

**Conceptualization:** Luchen Lin.

**Data curation:** Shuo Gu.

**Formal analysis:** Shuo Gu.

**Funding acquisition:** Jun Fu.

**Investigation:** Luchen Lin, Han He.

**Methodology:** Luchen Lin.

**Project administration:** Luchen Lin.

**Software:** Luchen Lin.

**Supervision:** Jun Fu.

**Visualization:** Zhenghong Chen.

**Writing – original draft:** Luchen Lin.

**Writing – review & editing:** Jun Fu.

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
