## [Decision Letter · Decision Letter 0]

Dear Dr. Fu,

Thank you for submitting your manuscript to PLOS ONE. After careful consideration, we feel that it has merit but does not fully meet PLOS ONE’s publication criteria as it currently stands. Therefore, we invite you to submit a revised version of the manuscript that addresses the points raised during the review process.

We look forward to receiving your revised manuscript.

Kind regards,

Omer Saleem, Ph.D.

Academic Editor

PLOS ONE

Journal Requirements:

When submitting your revision, we need you to address these additional requirements. 1. Please ensure that your manuscript meets PLOS ONE's style requirements, including those for file naming. The PLOS ONE style templates can be found at https://journals.plos.org/plosone/s/file?id=wjVg/PLOSOne_formatting_sample_main_body.pdf and https://journals.plos.org/plosone/s/file?id=ba62/PLOSOne_formatting_sample_title_authors_affiliations.pdf 2. Please note that PLOS ONE has specific guidelines on code sharing for submissions in which author-generated code underpins the findings in the manuscript. In these cases, we expect all author-generated code to be made available without restrictions upon publication of the work. Please review our guidelines at https://journals.plos.org/plosone/s/materials-and-software-sharing#loc-sharing-code and ensure that your code is shared in a way that follows best practice and facilitates reproducibility and reuse. 3. Thank you for stating the following financial disclosure: 1.      Natural Science Foundation of Hunan Provincial, grant number 2022JJ58025.2. Graduate Research Innovation Project of Hunan Provincial, grant number CX20240995.  Please state what role the funders took in the study.  If the funders had no role, please state: ""The funders had no role in study design, data collection and analysis, decision to publish, or preparation of the manuscript."" If this statement is not correct you must amend it as needed. Please include this amended Role of Funder statement in your cover letter; we will change the online submission form on your behalf.

Additional Editor Comments:

The paper has been reviewed by two reviewers. One has rendered a minor decision, while the other has rendered a major revisions. The reviewer comments aim to improve the quality of the paper. The authors are, therefore, requested to revise the paper as per the suggestions of the reviewers.

Reviewers' comments:

Reviewer's Responses to Questions

**Comments to the Author**

1. Is the manuscript technically sound, and do the data support the conclusions?

Reviewer #1: Yes

Reviewer #2: Partly

2. Has the statistical analysis been performed appropriately and rigorously?

Reviewer #1: Yes

Reviewer #2: No

3. Have the authors made all data underlying the findings in their manuscript fully available?

Reviewer #1: No

Reviewer #2: No

4. Is the manuscript presented in an intelligible fashion and written in standard English?

Reviewer #1: Yes

Reviewer #2: No

Reviewer #1: 1. Clarity in Problem Statement and Objectives:

The introduction should provide a clearer and more concise statement of the research problem. While the paper discusses the limitations of traditional PID control, the connection to the improved sparrow algorithm should be articulated more explicitly. The motivation for selecting this specific algorithm needs stronger justification.

2. Experimental Validation and Benchmarking:

The manuscript lacks a direct comparison of the proposed method with other state-of-the-art algorithms. It would be beneficial to include additional benchmark tests against other optimization techniques such as Genetic Algorithm (GA) or Particle Swarm Optimization (PSO) in practical diesel engine control scenarios.

3. Algorithm Implementation Details:

The section describing the improved sparrow algorithm lacks sufficient implementation details. The modifications to the standard sparrow search algorithm should be elaborated on with mathematical derivations or pseudo-code to enhance reproducibility.

4. Statistical Analysis of Results:

The performance improvement claims are primarily based on numerical results, but no statistical validation is presented. Confidence intervals, variance analysis, and standard deviations should be included to validate the consistency and robustness of the results.

5. Discussion on Computational Complexity:

The manuscript does not sufficiently address the computational complexity of the improved algorithm compared to standard SSA. The trade-off between computational cost and performance improvement should be discussed in more detail, especially for real-time applications.

6. Grammar and Syntax Corrections:

There are several grammatical errors throughout the text. For example, "However, the improved algorithm improves the control effect but also increases the computational complexity" should be rephrased to "However, while the improved algorithm enhances the control effect, it also increases computational complexity."

7. Consistency in Terminology:

The terms "fuzzy PID control" and "FPID" should be used consistently throughout the manuscript to avoid confusion.

8. Figure and Table Captions:

Some figure captions lack detailed descriptions. For example, Figure 9 should explicitly state what the plotted data represent.

9. Units and Notations:

Ensure consistency in units (e.g., "RPM" vs. "r/min"). Some values are given in different formats across sections.

10. Reference Formatting:

Some citations are not formatted according to PLOS ONE guidelines. Ensure uniformity in references, particularly in citing author names and publication years.

11. Clarification on Experimental Conditions:

The manuscript does not specify whether real-world diesel engine tests were conducted or if all results are from simulations. This should be explicitly stated in the methodology.

12. Equation Formatting and Explanation:

Some equations lack proper explanation, such as Equation (4). The meaning of variables should be explicitly defined.

13. Abstract Refinement:

The abstract should be more concise and should highlight the key contributions in a clearer manner, including the percentage improvements achieved.

14. Improved Transition Between Sections:

Some sections transition abruptly. For example, the shift from theoretical modeling to algorithm description should be smoother, with better contextual bridging.

15. Addressing Limitations and Future Work:

The paper should include a more detailed discussion on the limitations of the proposed method and potential areas for future research, such as real-time implementation challenges or hybrid optimization strategies.

Overall, the manuscript presents an interesting approach to optimizing diesel engine speed control using an improved sparrow search algorithm. However, several aspects need further refinement, particularly in experimental validation, statistical analysis, and computational efficiency discussions. Implementing these revisions will significantly enhance the paper's quality and impact.

Reviewer #2: Concerns

1. In the following sentence [6] will be before full stop.

Traditional PID controllers cannot perform online parameter tuning, and in complex systems such as diesel engines, which are nonlinear and time-varying, their control effects are often unsatisfactory. [6]

2. In the introduction section, performance of cited studies could be added briefly.

3. “ELM” is mentioned in the below paragraph but what it stands for is missing:

Zhang [27] improved the sparrow algorithm for an ELM neural network to propose a method for the inversion of soil parameters for underground space development and the prediction of deformation of underground structures.

4. “RPM” is written multiple times but what it stands for is not described anywhere.

5. Reference of basic equations 1, 2, 4, 5, 11 should be added.

6. In the following equation d-phi is not explained:

7. In the description of following equation, some variables are not explained:

8. Qw is not explained for the following equation

9. a_g is written in paragraph, whereas the equation is

10. Is it “Engine modle” in Fig. 3?

11. It should be explained clearly that why sparrow algorithm is specifically considered for optimizing the algorithm.

12. In eq. 14, ST is not explained. Second, the link between R2 with X is missing.

13. Derivation of eq. 16 lacks clarity.

14. The derivation of eq. 17 and 18 must be added. How the authors finally reached at these equations?

15. Fig. 9 is written twice with same title.

16. It would be interesting to see how the cylinder pressure behaves under different loads.

17. Since authors propose improved SSA, I strongly recommend that its flow chart should be added in the manuscript.

18. It should mentioned that how would the system behave under transient operating conditions?

**Do you want your identity to be public for this peer review?** For information about this choice, including consent withdrawal, please see our Privacy Policy

Reviewer #1: No

Reviewer #2: No

---

## [Author Response · Author response to Decision Letter 1]

19 May 2025

Dear Editor and Reviewers

We sincerely thank the editor and all reviewers for their valuable feedback that we have used to improve the quality of our manuscript. The reviewer comments are laid out below in italicized font and specific concerns have been numbered. Our response is given in normal font and changes to the manuscript are given in the red text.

Q1: Clarity in Problem Statement and Objectives:The introduction should provide a clearer and more concise statement of the research problem. While the paper discusses the limitations of traditional PID control, the connection to the improved sparrow algorithm should be articulated more explicitly. The motivation for selecting this specific algorithm needs stronger justification.

A1: Thanks for your suggestion. In response to the above problems, we have made changes in the article as follows Line 65 shows that the Sparrow algorithm has higher accuracy and faster response speed than other algorithms, and three related literature articles are added at the end of the paragraph to show that the algorithm is superior to other algorithms in the process of use. In line 73, the case literature is analyzed and summarized, and the algorithm improvement direction is proposed by considering how to combine the application of the algorithm with the speed control of diesel engine.

The specific changes are as follows

In recent years, the sparse search algorithm (SSA), an emerging intelligent optimization algorithm[24], has performed well in a variety of optimization problems owing to its simplicity, efficiency, and good adaptability[25,26,27].

The above study shows that the SSA performs well for various engineering problems.

Above scholars have addressed the inherent defects of the pigeon-inspired algorithm, such as population initialization imbalance, slow convergence speed, and tendency to get trapped in local optima, by proposing different solutions. In the diesel engine speed control system, the optimization accuracy of control parameters directly affects the speed regulation performance, imposing stricter requirements on the global search capability and convergence accuracy of the optimization algorithm.

This study aims to propose an improved sparrow algorithm for the problem of diesel speed control under load variation. It is applied to the parameter optimization of a fuzzy PID controller, achieving stable control of the diesel engine speed under load changes by optimizing the proportional and quantization factors of the fuzzy PID.

Therefore, this study proposes an improved Squirrel Search algorithm to address the problem of the algorithm easily falling into local optima due to the high precision requirements for parameter adjustment in diesel engine speed control, and achieves stable control of the diesel engine speed under load changes by optimizing the proportion factor and quantization factor of the FPID.

Q2: Experimental Validation and Benchmarking:The manuscript lacks a direct comparison of the proposed method with other state-of-the-art algorithms. It would be beneficial to include additional benchmark tests against other optimization techniques such as Genetic Algorithm (GA) or Particle Swarm Optimization (PSO) in practical diesel engine control scenarios.

A2: We sincerely thank you for your valuable suggestions and believe they will be very helpful in improving the quality of the manuscripts. Therefore, we added the particle swarm optimization algorithm (PSO) to the comparative test and presented the experimental results in Figure 8.

Q3: Algorithm Implementation Details:The section describing the improved sparrow algorithm lacks sufficient implementation details. The modifications to the standard sparrow search algorithm should be elaborated on with mathematical derivations or pseudo-code to enhance reproducibility.

A3: Thanks for your suggestion. We had supplemented the algorithmic implementation process by adding the pseudo-code of the ISSA algorithm at line 287 in the manuscript to enhance the reproducibility of the algorithm.

The pseudo-code implementation process of improving the sparrow algorithm is as follows:

Algorithm 1 The ISSA Algorithm

Input

n: the number of sparrows

PD: the number of the producer sparrows

SD: the number of the sparrows which perceive danger

Itermax: the total number of iterations

R2: the alarm value

output

Xbest: the global optimal position

fbest: the optimal solution

1: Using (21) to Initialize a population of n sparrows

2:whiel(t<Itermax)

3: Calculating the fitness values of the individuals;

4: Ranking the fitness values and finding the current best individual and the current worst individual;

5: For I =1: PD

6: Using (20) to update the producer position;

7: end for

8: For I =(PD+1): n

9: Using (21) to update the scrounger position;

10: end for

11: For I = 1: SD

12: Using (22) to update the Scout Position;

13: end for

14: Obtaining the current new position;

15� if (the new position is better than the best position)

16: Replacing the best position with the new position

17: end if

18: t=t+1;

19: end while

20: return Xbest, fbest;

Q4: Statistical Analysis of Results:The performance improvement claims are primarily based on numerical results, but no statistical validation is presented. Confidence intervals, variance analysis, and standard deviations should be included to validate the consistency and robustness of the results.

A4: Thank you for your valuable feedback, the issue you mentioned is very crucial. The improvement of algorithm performance needs to include Confidence intervals, variance analysis, and standard deviations. Therefore, in line 291 of the text and Table 2, it is pointed out that all tests were run independently 30 times to eliminate the influence of random numbers, with 1000 iterations and a population size of 100 for the algorithm. Reference “A novel inversion approach for seepage parameter of concrete face rockfill dams based on an enhanced sparrow search algorithm” is used for algorithm validation, using the optimal value, average value, and standard deviation as the basis for testing. And combine the three tables into one table

The specific changes are as follows

To validate the performance and optimization results of the improved SSA, four benchmark functions with significant optimization features were selected. These functions were tested in 30 dimensions with a population size of 100, and 30 tests were performed to evaluate the improved algorithm, as listed in Table 2.

To verify the performance and optimization results of the improved SSA, four benchmark functions with significant optimization characteristics were selected. These functions were independently run 30 times to avoid the influence of random numbers, with 1000 iterations set and a population size of 50. he specific dimensions, ranges, and extreme values of the test functions are detailed in Table 2.

Table 3. Optimal values of test function search results

Function ISSA SSA GWO PSO

F1 0 4.82E−98 5.86E-86 3.91E-3

F2 1.89E-22 5.44E-13 25.17 2.30

F3 0 0 0 3.97

F4 0 0 0.01 0.17

Table 4. Average values of the test function search results

Function ISSA SSA GWO PSO

F1 0 2.55E-55 1.39e-85 0.21

F2 1.07E-07 1.79E-02 25.19 7.74

F3 0 1.02e-12 0 4.75

F4 0 1.66e-13 0.02 0.21

Table 3. Comparison of test results of different algorithms.

Indicator Function ISSA SSA GWO PSO

Beast value F1 0 4.82E−98 5.86E-86 3.91E-3

F2 3.31E-07 1.03E-05 5.42E-05 0.017

F3 0 0 0 3.97

F4 0 0 0.01 0.17

Average values F1 0 2.5E-52 8.07E-85 0.047

F2 6.85E-06 4.16E-04 2.84E-04 0.104

F3 0 1.51E-14 0 7.721

F4 0 1.87E-16 0.023 0.154

Strand deviation F1 0 1.38E-51 1.66587E-84 0.057

F2 6.14E-06 4.83E-04 1.57E-04 0.067

F3 0 4.70E-14 0 1.919

F4 0 1.87E-16 0.023 0.154

In this study, four optimization algorithms were compared, namely, particle swarm optimization (PSO), gray wolf optimization (GWO), the Sparrow algorithm (SSA), and the improved Sparrow algorithm (ISSA). Figure 4 and Table 3 show the convergence curves of these algorithms on the four test functions, from which it can be seen that the ISSA algorithm exhibits the best performance among all test functions. Table 4 shows the average values after 30 tests of the algorithm, demonstrating the high stability and accuracy of the ISSA algorithm compared with the four other algorithms. The results indicate that the ISSA algorithm exhibits stronger stability among the four test functions, suggesting that the algorithm can overcome local optimum issues and obtain the best test results.

In this study, four optimization algorithms were compared, namely, particle swarm optimization (PSO), gray wolf optimization (GWO), the Sparrow algorithm (SSA), and the improved Sparrow algorithm (ISSA). Figure 4 show the convergence curves of these algorithms on the four test functions, from which it can be seen that the ISSA algorithm exhibits the best performance among all test functions. Table 3 shows the optimal values, average values, and standard deviations of four algorithms after 30 independent runs. It can be found that the ISSA algorithm performs the best performance under three indices, and it can find the most extreme values in the F1, F3, F4 test functions, and is closer to the extreme value than the other algorithms in the F2 test function. At the same time, both the average values and standard deviations after 30 runs are better than those of the other algorithms.

Q5: Discussion on Computational Complexity:The manuscript does not sufficiently address the computational complexity of the improved algorithm compared to standard SSA. The trade-off between computational cost and performance improvement should be discussed in more detail, especially for real-time applications.

A5: Thanks for your valuable comments on improving the computational complexity of the ISSA and its impact on real-time applications. Here is my reply to this question.

In the original algorithm, the computational complexity involves the roles of explorers, followers, and sentinels, fitness evaluation, sorting, and selection decisions. The main improvements in the ISSA algorithm include circular chaotic mapping initialization, dynamic search factor, sine-cosine strategy, and Lévy flight strategy. These improvements do not add more operation steps or structures, so overall, the ISSA maintains the same asymptotic complexity as SSA, but the ISSA algorithm has faster convergence speed, which can be demonstrated in Figure 8. For real-time applications like diesel engines, the algorithm is mainly used for offline optimization and online fine-tuning of fuzzy rules.

Q6:Grammar and Syntax Corrections: There are several grammatical errors throughout the text. For example, "However, the improved algorithm improves the control effect but also increases the computational complexity" should be rephrased to "However, while the improved algorithm enhances the control effect, it also increases computer

A6: Thanks for your valuable comments. We carefully check and revise the errors you raised, and at the same time carefully check the article to correct the grammatical errors in the text, and attach a test report.

However, the improved algorithm improves the control effect but also increases the computational complexity

However, while the improved algorithm enhance the control effect, it also increase the compute

Q7: Consistency in Terminology: The terms "fuzzy PID control" and "FPID" should be used consistently throughout the manuscript to avoid confusion.

A7: Thanks for your valuable feedback. This is an imprecise error which I have corrected. Fuzzy PID is replaced by FPID.

Q8: Figure and Table Captions: Some figure captions lack detailed descriptions. For example, Figure 9 should explicitly state what the plotted data represent.

A8: Thanks for your valuable feedback. In my manuscript, Figure 9 appeared twice, and this error has been corrected. Figure 9 represents the response curve of the diesel engine from idle to the target speed of 2000 RPM. Figure 10 represents the random load under load changes.

具体修改如下�Figure 10. Load Variation Figure 10.different load 6-10

Q9: Units and Notations: Ensure consistency in units (e.g., "RPM" vs. "r/min"). Some values are given in different formats across sections.

A9: Thanks for your valuable feedback. I have unified the units and replaced them with RPM.

Q10: Reference Formatting: Some citations are not formatted according to PLOS ONE guidelines. Ensure uniformity in references, particularly in citing author names and publication years.

A10: Thank you for your valuable comments. I have carefully revised the references according to the requirements of your journal, as follows:

[1]Sujesh G ,Ramesh S .Modeling and control of diesel engines: A systematic review[J].Alexandria Engineering Journal,2018,57(4):4033-4048.

[2]Yeom J ,Jung S ,Yoon J .An experimental study on the application of oxygenated fuel to diesel engines[J].Fuel,2019,248262-277.

[3]Frank L H I ,Bolan L ,Fanshuo L , et al.Review of Artificial Intelligent Algorithms for Engine Performance, Control, and Diagnosis[J].Energies,2023,16(3):1206-1206.

[4]Åström K J, Hägglund T. PID control[J]. IEEE Control Systems Magazine, 2006, 1066.

[5]Borase R P, Maghade D K, Sondkar S Y, et al. A review of PID control, tuning methods and applications[J]. International Journal of Dynamics and Control, 2021, 9: 818-827.

[6]Somefun O A, Akingbade K, Dahunsi F. The dilemma of PID tuning[J]. Annual Reviews in Control, 2021, 52: 65-74.

[7]Padula F, Visioli A. Tuning rules for optimal PID and fractional-order PID controllers[J]. Journal of process control, 2011, 21(1): 69-81.

[8]Zhao C, Guo L. PID controller design for second order nonlinear uncertain systems[J]. Science China Information Sciences, 2017, 60: 1-13.

[9]Yu W, Rosen J. Neural PID control of robot manipulators with application to an upper limb exoskeleton[J]. IEEE Transactions on cybernetics, 2013, 43(2): 673-684.

[10]Xu J, Xiong Z, Bhattacharyya S P. PIDNet: A real-time semantic segmentation network inspired by PID controllers[C]. Proceedings of the IEEE/CVF conference on computer vision and pattern recognition. 2023: 19529-19539.

[11]Sahib M A, Ahmed B S. A new multiobjective performance criterion used in PID tuning optimization algorithms[J]. Journal of advanced research, 2016, 7(1): 125-134.

[12]Mishra P, Kumar V, Rana K P S. A fractional order fuzzy PID controller for binary distillation column control[J]. Expert Systems with Applications, 2015, 42(22): 8533-8549.

[13]Najariyan M, Zhao Y. Granular fuzzy PID controller[J]. Expert Systems with Applications, 2021, 167: 114182.

[14]Cong S, Liang Y. PID-like neural network nonlinear adaptive control for uncertain multivariable motion control systems[J]. IEEE Transactions on Industrial Electronics, 2009, 56(10): 3872-3879.3

[15]Zhao Y, Du X, Xia G, et al. A novel algorithm for wavelet neural networks with application to enhanced PID controller design[J]. Neurocomputing, 2015, 158: 257-267.

[16]Zeng B ,Shen Q ,Wang G , et al. Research on Diesel Engine Speed Control Based on Improved Salp Algorithm[J].Processes,2023,11(11):

[17]Zhu Cunxi. Research on Speed Control Strategy of Marine Medium - speed Engine Based on Genetic Algorithm [D]. Harbin Engineering University, 2023.

[18]Zhuo H ,Weihao G ,Kege Z , et al.Optimization of PID control parameters for marine dual-fuel engine using improved particle swarm algorithm[J].Scientific Reports,2024,14(1):12681-12681.

[19]Ding S L, He S F, Tu B Q, et al. Model‐Based Control with Active Disturbance Rejection Algorithm for a Diesel Engine[J]. Complexity, 2023.

[20]Xiang D ,Ying H ,Yanwu G , et al. Fuzzy-PID Speed Control of Diesel Engine Based on Load Estimation [J]. SAE International Journal of Engines, 2015, 8 (4): 1669-1677.

[21]Jie Y ,Fu Y S ,Bo-Chiuan C , et al.Application of Adaptive Idle Speed Control on V2 Engine[J].SAE International Journal of Engines,2015,9(1):458-465.

[22]Zhangmiaoge L ,Zhouxiao L ,Jianzhao L , et al. Thermal management with fast temperature convergence based on optimized fuzzy PID algorithm for electric vehicle battery [J]. Applied Energy, 2023, 352

[23]Azizi M ,Ejlali G R ,Ghasemi M A S , et al.Upgraded Whal

---

## [Decision Letter · Decision Letter 1]

Dear Dr. Fu,

Thank you for submitting your manuscript to PLOS ONE. After careful consideration, we feel that it has merit but does not fully meet PLOS ONE’s publication criteria as it currently stands. Therefore, we invite you to submit a revised version of the manuscript that addresses the points raised during the review process.

We look forward to receiving your revised manuscript.

Kind regards,

Mohamed Yacin Sikkandar

Academic Editor

PLOS ONE

Journal Requirements:

Reviewers' comments:

Reviewer's Responses to Questions

**Comments to the Author**

Reviewer #1: All comments have been addressed

Reviewer #3: All comments have been addressed

2. Is the manuscript technically sound, and do the data support the conclusions?

Reviewer #1: Yes

Reviewer #3: No

3. Has the statistical analysis been performed appropriately and rigorously?

Reviewer #1: No

Reviewer #3: Yes

4. Have the authors made all data underlying the findings in their manuscript fully available?

Reviewer #1: No

Reviewer #3: Yes

5. Is the manuscript presented in an intelligible fashion and written in standard English?

Reviewer #1: Yes

Reviewer #3: Yes

Reviewer #1: 1. Although the diesel engine model validation is mentioned, provide more explicit details about the exact conditions under which experimental validation was conducted. Include specific testing conditions, environment details, and numerical validation metrics to ensure replicability and validity of results.

2. The manuscript should discuss in-depth the computational complexity implications of integrating ISSA with fuzzy PID. Specifically, quantify the computational load and the practical feasibility of this improved algorithm for real-time implementation.

3. Provide a more detailed and stepwise explanation of the optimization strategy integrating Circle Chaotic Mapping, dynamic search factors, sine and cosine strategies, and Levy flight strategies. Currently, these are described independently; an integrated and coherent discussion will strengthen understanding and methodological rigor.

4. Include a dedicated section discussing robustness analysis. Address scenarios with varying magnitudes of disturbances and quantify robustness using established metrics (e.g., Integral of Time-weighted Absolute Error (ITAE), Integral of Absolute Error (IAE), etc.) to demonstrate the real-world applicability of the proposed control strategy.

5. Clearly demonstrate the comparison between the improved algorithm and existing methods (SSA, PSO, GA, etc.) with explicit numerical values and comprehensive statistical analysis (e.g., standard deviation, confidence intervals, statistical significance testing) to confirm the claimed superiority.

6. Correct minor formatting inconsistencies in the manuscript, particularly in equations and variable definitions, to maintain uniformity and clarity throughout the document.

7. Enhance the visual quality and clarity of all graphs and diagrams. Specifically, improve axis labeling, legends, and ensure resolution is sufficient for readability.

8. Clearly define all abbreviations upon first use and provide consistent terminology, especially for fuzzy PID parameters and ISSA algorithm components.

9. Conduct a thorough English language editing to fix minor grammatical errors and awkward phrasing, improving readability and professionalism.

Reviewer #3: The revised manuscript entitled "Research on the speed fluctuation control of diesel engine under load changes via an improved sparrow algorithm"

my recommendation is Accept

**Do you want your identity to be public for this peer review?** For information about this choice, including consent withdrawal, please see our Privacy Policy

Reviewer #1: No

Reviewer #3: No

---

## [Author Response · Author response to Decision Letter 2]

28 Jun 2025

Dear Editor and Reviewers

We sincerely thank the editor and all reviewers for their valuable feedback that we have used to improve the quality of our manuscript.

1.Although the diesel engine model validation is mentioned, provide more explicit details about the exact conditions under which experimental validation was conducted. Include specific testing conditions, environment details, and numerical validation metrics to ensure replicability and validity of results.

A1:Thank you for your valuable comments regarding the model validation section.We added the picture of the diesel engine test bench as shown below, and all the experimental data were obtained from the bench data.

This study focuses on small diesel engines by establishing a diesel engine submodel (cylinder, intake and exhaust, and crankshaft power model) to accurately reflect the operating characteristics of the diesel engine. Through the experimental setup shown in the figure, accurate operating data of the diesel engine can be obtained and the simulation model can be verified to ensure the model error accuracy is less than 10%, providing a reliable simulation model for subsequent algorithm development. The basic parameters of the diesel engine used in this study are shown in Table 1.

This study focuses on small diesel engines by establishing a diesel engine submodel (cylinder, intake and exhaust, and crankshaft power model) to accurately reflect the operating characteristics of the diesel engine. Through the experimental setup shown in the figure 1, accurate operating data of the diesel engine can be obtained and the simulation model can be verified to ensure the model error accuracy is less than 10%, providing a reliable simulation model for subsequent algorithm development. The basic parameters of the diesel engine used in this study are shown in Table 1.

Figure 1.Engine test bench

2.The manuscript should discuss in-depth the computational complexity implications of integrating ISSA with fuzzy PID. Specifically, quantify the computational load and the practical feasibility of this improved algorithm for real-time implementation.

A2:Thank you for your suggestions. While an in-depth analysis of the computational complexity of ISSA combined with fuzzy PID is crucial for real-time applications, this study primarily focuses on investigating how the improved algorithm optimizes the controller's performance. Since ISSA serves as an offline optimization tool, real-time performance was not considered in this work. Therefore, we have acknowledged this limitation in the outlook section and will prioritize a quantitative analysis of the algorithm's computational efficiency in follow-up research.

3.Provide a more detailed and stepwise explanation of the optimization strategy integrating Circle Chaotic Mapping, dynamic search factors, sine and cosine strategies, and Levy flight strategies. Currently, these are described independently; an integrated and coherent discussion will strengthen understanding and methodological rigor.

A3:Thank you for your suggestions.In response to the previous lack of comprehensive description regarding the improvement strategies for the Sparrow Search Algorithm, we have revised our manuscript to explicitly clarify the specific problems addressed and advantages offered by each of the three optimization strategies before introducing the improved algorithm. The combined implementation of these strategies provides a holistic solution to the algorithm's limitations, with detailed modifications as follows:

Although the sparrow algorithm has strong global search ability, it is not uniformly distributed in the high-dimensional and complex parameter space, the convergence speed is slow, and there is a risk of a local optimal solution, which may lead to optimization results of the control parameters that are not globally optimal and affect the accuracy and stability of the control of the diesel engine speed. This paper proposes the following optimization strategy to address the above problems.

Although the Sparrow Search Algorithm (SSA) possesses strong global search capabilities, it demonstrates uneven distribution in high-dimensional complex parameter spaces, slow convergence speed, and risks of falling into local optima, which may lead to non-globally optimal control parameter optimization results, thereby affecting the accuracy and stability of diesel engine speed control. To address these issues, this paper proposes three combined optimization strategies to improve the aforementioned shortcomings: introducing an improved Circle chaotic map to enhance the random distribution of the initial population and avoid population unevenness; adopting a dynamic factor during the exploration phase to initially expand and then progressively narrow the search range; utilizing a sine-cosine strategy that leverages the periodicity, fluctuation, and symmetry of sine and cosine functions to balance the algorithm's global search and local search capabilities; and employing a Levy flight strategy that combines long-step jumps with frequent short-step movements, enabling the search process to achieve both local refined exploration and globally efficient search.

4.Include a dedicated section discussing robustness analysis. Address scenarios with varying magnitudes of disturbances and quantify robustness using established metrics (e.g., Integral of Time-weighted Absolute Error (ITAE), Integral of Absolute Error (IAE), etc.) to demonstrate the real-world applicability of the proposed control strategy.

A4:Thank you for your valuable feedback. Robustness analysis is indeed crucial, and in this study, we have demonstrated the superiority of the proposed algorithm in diesel engine parameter optimization through the fitness function shown in Figure 10. Figures 11 and Table 4 provide a detailed analysis of the control performance at 2000 RPM, while Figure 12 illustrates the speed fluctuations under load variations, effectively showcasing the enhanced stability achieved through parameter optimization.

5.Clearly demonstrate the comparison between the improved algorithm and existing methods (SSA, PSO, GA, etc.) with explicit numerical values and comprehensive statistical analysis (e.g., standard deviation, confidence intervals, statistical significance testing) to confirm the claimed superiority.

A5:We sincerely appreciate the valuable suggestions provided. As referenced in [ ], we have conducted systematic comparisons between the improved algorithm and existing methods (including SSA, PSO, and GA) using three key evaluation metrics: (1) optimal fitness value (indicating algorithm accuracy), (2) average of 30 runs (reflecting overall performance), and (3) standard deviation (demonstrating stability). The current experimental results have effectively validated the enhancement effects of our proposed algorithm. In future studies, we plan to incorporate more comprehensive statistical analysis methods to further improve our comparative experiments. We would like to express our sincere gratitude to the reviewers for their constructive comments on this important issue

6.Correct minor formatting inconsistencies in the manuscript, particularly in equations and variable definitions, to maintain uniformity and clarity throughout the document.

A6:Thank you for your valuable suggestions. Through our careful proofreading, we identified duplicate variable definitions in the equations and have made corresponding revisions.

is the specific enthalpy of the workmass at the inlet and exhaust valves

is the specific enthalpy of the workmass at the inlet and exhaust valves

7.Enhance the visual quality and clarity of all graphs and diagrams. Specifically, improve axis labeling, legends, and ensure resolution is sufficient for readability.

A7:Thank you for your valuable comments, we have modified the quality of the chart, optimized the data pictures, including unified fonts, complete axis annotation and improved the quality of the picture to ensure clarity, thank you again for your valuable comments.

8.Clearly define all abbreviations upon first use and provide consistent terminology, especially for fuzzy PID parameters and ISSA algorithm components.

A8:Thank you for your valuable suggestions. Regarding the terminology standardization issues you raised, the first occurrence of "fuzzy Proportional Integral Derivative" is uniformly named as FPID, and the "Improved Sparrow Search Algorithm" is specified as ISSA.

9.Conduct a thorough English language editing to fix minor grammatical errors and awkward phrasing, improving readability and professionalism.

A9:Thanks to your valuable comments, we have once again corrected the grammar issues and made changes to individual vocabulary as follows:

1�The response time reduced by 0.47 s and the maximum overshoot reduced by 98.43%.

The response time was reduced by 0.47 s and the maximum overshoot was reduced by 98.43%.

2� is the prevalve mass pressure (bar); and is the prevale mass-specific volume.

is the prevalvence mass pressure (bar); and is the prevalent mass-specific volume.

---

## [Editor Report · Decision Letter 2]

Research on the speed fluctuation control of diesel engine under load changes via an improved sparrow algorithm

PONE-D-25-07096R2

Dear Dr. Fu,

We’re pleased to inform you that your manuscript has been judged scientifically suitable for publication and will be formally accepted for publication once it meets all outstanding technical requirements.

Kind regards,

Mohamed Yacin Sikkandar

Academic Editor

PLOS ONE